# Pharmacological Potential of Fungal Endophytes Associated with Medicinal Plants: A Review

**DOI:** 10.3390/jof7020147

**Published:** 2021-02-17

**Authors:** Bartholomew Saanu Adeleke, Olubukola Oluranti Babalola

**Affiliations:** Food Security and Safety Niche Area, Faculty of Natural and Agricultural Sciences, North-West University, Private Bag X2046, Mmabatho 2735, South Africa; microbade@yahoo.com

**Keywords:** antibiotics, antimicrobial, fungal metabolites, medicinal plants, pharmacology

## Abstract

Endophytic microbes are microorganisms that colonize the intracellular spaces within the plant tissues without exerting any adverse or pathological effects. Currently, the world population is facing devastating chronic diseases that affect humans. The resistance of pathogens to commercial antibiotics is increasing, thus limiting the therapeutic potential and effectiveness of antibiotics. Consequently, the need to search for novel, affordable and nontoxic natural bioactive compounds from endophytic fungi in developing new drugs with multifunction mechanisms to meet human needs is essential. Fungal endophytes produce invaluable bioactive metabolic compounds beneficial to humans with antimicrobial, anticancer, antidiabetic, anti-inflammatory, antitumor properties, etc. Some of these bioactive compounds include pestacin, taxol, camptothecin, ergoflavin, podophyllotoxin, benzopyran, isopestacin, phloroglucinol, tetrahydroxy-1-methylxanthone, salidroside, borneol, dibenzofurane, methyl peniphenone, lipopeptide, peniphenone etc. Despite the aforementioned importance of endophytic fungal metabolites, less information is available on their exploration and pharmacological importance. Therefore, in this review, we shall elucidate the fungal bioactive metabolites from medicinal plants and their pharmacological potential.

## 1. Introduction

Medicinal plants harbor diverse populations of microbial domains. Their interdependence in the synthesis of bioactive compounds as a therapy to confront the emergence of new drug-resistant pathogens has increased the quest for an alternative to chemosynthetic drugs for treating human diseases [1,2,3]. Despite the tiny nature of fungi, for example, *Penicillium notatum* has been used as an invaluable source of novel metabolite compounds and broad-spectrum antibiotics, as credited to Alexander Fleming in the discovery of penicillin [4]. Since then, this antibiotic has continued to dominate the market and has also awakened the consciousness of scientists in the search for bioactive metabolites from fungi inhabiting the soil (rhizosphere) or plant endosphere [5,6]. 

The endosphere depicts the inner part of the plant with different microorganisms inhabiting the zone, completing their life span without exercising pathological consequences on the host plants [7,8]. Fossil records in the study of plant microbial ecology have shown the prehistorical interaction of fungal endophytes with plants; most fungal endophytes interacting with plants originated when higher plants were first seen on Earth, as documented in [9]. Fungal endophytes are symbiotically associated with their host plants; their mutual interdependence enabled them to confer beneficial effects and support each other [10]. Plants secrete exudates that supply nutrients to the endomycorrhiza in the rhizosphere and other endophytes infiltrating the root tissues; in return, the fungal endophytes chiefly support their host plants to survive under different environmental stresses by the secretion of stress-adaptor metabolites [11,12,13].

To date, reports have shown that 7% of 1.5 million fungal species have been identified; although recent findings using next-generation sequencing have revealed that between 3.5 and 5.1 million fungal species exist on earth [14], as opposed to earlier reports in 2017 by Hawksworth and Lücking [15] that about 2.2 to 3.1 million fungal species exist worldwide. Fungal endophytes are thus dominating the ecosystem and can be promising as a source of different bioactive compounds.

It is expected that despite the multidisciplinary research approached in the study of endophytes and its various bioactive compounds, the discipline is still in its infancy, and little success has been recorded in the synthesis of endophytic fungal bioactive compounds for commercial purposes. Therefore, the biotechnological application of fungal endophytes in agriculture and industries in the derivation of salient natural plant products became necessary. The discovery of new metabolic compounds from medicinal plants and associated fungi has been documented in several research findings [16,17,18]. Currently, *Pelargonium sidoides* has been recognized as the most invaluable medicinal plant used traditionally in primary health care in South Africa and research on the antibacterial properties of the diverse endophytic fungal extract from *Pelargonium sidoides* DC has been documented [17]. The resulting potentials of this plant and other medicinal plants in the synthesis of bioactive compounds have been documented [19]. Likewise considered, but less studied, are the bioactive compounds that are synthesized by the endophytic fungi. Hence, critical elucidation on endophyte ecology, bioactive components, and the biotransformation of fermentable substrates in culturing and biomass production is paramount. Therefore, in this review, the various bioactive compounds from endophytic fungi associated with medicinal plants and their pharmacological importance shall be discussed. 

## 2. Bioactive Compounds from Medicinal Plants 

In nature, different medicinal plants are found in the ecosystem as a source of curative substances for various diseases, and are employed in modern therapeutics as an alternative to chemical drugs [20]. In developing countries, approximately 80% of people rely on herbal medicines as their sole primary healthcare. Studies have revealed that about 51% of new drugs developed between 1981 and 2014 were naturally synthesized from plants, as they have been experimentally tested against some infectious diseases and for treating cancer [21,22]. Globally, the trend of medicinal plant usage is increasing based on the high demand for herbal drugs and pharmaceuticals. However, the cultivation of medicinal plants is faced with many challenges arising from overharvesting without replanting, ecological distortion, natural anthropogenic activities, and the destruction of habitat by pest infestation, thus reducing the plant population and its exploration for medicinal use [16]. 

In reality, the exploitation of plant bioactive products can be infeasible due to the small amount of product accumulation in plants, long maturation time, and difficulty in the recovery of plant metabolites. Nevertheless, to maximally explore bioactive compounds from plants, it is imperative to devise an alternative means of scaling up the cultivation of medicinal plants to meet demand through the total chemical synthesis, heterologous production, tissue culture, semisynthesis, microbial synthesis or plant synthesized natural products or exploitation of endophytic microbes (fungi) in the synthesis of the corresponding bioactive compounds as their hosts [23]. Research has established that approximately 18% of plant-derived metabolites can as well be derived from its associated fungi [24]. For example, similar bioactive compound, Taxol has been derived from a medicinal plant, *Taxus*, and its associated endophytic fungi *Taxomyces andreanae* [25]. Hence, the Taxol from *T. andreanae* can preferably stand promising over its host plant due to ease of exploration by fermentation and production [26]. Furthermore, studies in the literature on some medicinal plants and endophytes in the synthesis of bioactive compounds with more emphasis on plant species and organs as sources of bioactive metabolites have been reported [27,28]. 

## 3. Pharmacological Effects of Bioactive Compounds of Fungal Metabolites 

Endophytic fungi potentially produce novel bioactive compounds. Suitable media, growth parameters and nutrient limitation should be explored to gain insight into fungal metabolism to discover novel pharmaceutical products [29]. Hence, the pharmacological properties of major bioactive compounds synthesized from endophytic fungi and significant effects are represented in Table 1 and further highlighted below:

### 3.1. Antifungal

Recent research findings into fungal endophytes as a source of antimicrobial compounds have provided alternative means in overcoming drug resistance by pathogens, the inefficiency of antibiotic activity against bacterial species, and a limited amount of antimicrobial production [41]. The antibacterial, antifungal, antiprotozoan, and antiviral activities of antimicrobial compounds synthesized by endophytic fungi have helped prevent various disease conditions affecting living organisms [17,42]. 

Examples of antimicrobial compounds produced by endophytic fungi include clavatol, chaetomugilin D, guignardic acid, colletotric acid, viridicatol, 7-amino-4-methylcoumarin, altersolanol A, 2-hydroxyl-6 methyl benzoic acid, enfumafungin, xylarenone B, jesterone, pestacin, hydroxy-jesterone, rutin, fusapyridon A, hypericin, phomopsin A, isopestacin, xylarenic acid, fusaripeptide A, xylarenone A, javanicin, Z-roquefortine C, penitrem A, penijanthine A, fusarithioamide A, pestalone, cryptocandin cryptocin, ecomycins, pseudomycins, pestaloside and pestalopyrone [43]. The isolation of *Streptomyces* spp. TQR12-4 from Elite *Citrus nobilis* fruit with broad-spectrum antimicrobial activity against fungi pathogens *Colletotrichum truncatum*, *Geotrichum candidum*, *F. oxysporum*, and *F. udum* has been reported [44].

Some bioactive compounds such as triterpenoids, sesquiterpenes, and diterpenoids produced from endophytic fungi account for their antimicrobial activity against some fungal and bacterial pathogens [45]. In Chinese herbal medicines, diverse fungal endophyte communities isolated from the roots of *Paris polyphylla* have traditionally been employed. Similarly, the colletotric acid (C_28_H_28_O_10_) antimicrobial substance produced by the fungal endophyte *Colletotrichum gloesporoides* from the Chinese medicinal plant *Artemisis mongolica* has potentially displayed antimicrobial properties [46].

Antimicrobial compounds are relatively low in molecular weight, and even at a low concentration (0.5–8.0 mg/mL), they exhibit high activity against pathogenic microorganisms [47]. Endophytes undergo indirect mechanisms by secreting biocides inhibitory to plant pathogens [48]. This mechanism is believed to be powered by the bioactive metabolites produced by the endophytes. Different antimicrobial compounds such as peptides, phenols, terpenoids, steroids, alkaloids, flavonoids, and quinine, produced by endophytes, have been reported [49]. 

Based on human perception, antimicrobial compounds can be employed in diverse ways not only as drugs but also in food processing and preservation. Some antimicrobials exhibited lethal effects on pathogenic or spoilage microorganisms causing food poisoning or intoxication. The production of antimicrobial compounds from endophytic fungus *Armillaria mellea* with high bioactivity against Gram-positive bacterial and fungal pathogens has been reported [50]. Furthermore, some endophytic fungi in the genus *Xylaria* produce certain bioactive compounds with antifungal activity. Examples of such compounds include sordaricin, 1, 8-dihydroxynaphthol 1-O-a-glucopyranoside, mellisol, and multiplolides A and B. Sordarcin and multiplolides A and B, which display antifungal activity against *Candida albicans* while 1, 8-dihydroxynaphthol 1-O-a-glucopyranoside and mellisol with antifungal activity against herpes simplex virus-type 1 are also known. The *Xylaria* spp. YX-28 isolated from *Gingko biloba* produced bioactive compound 7-amino-4-methylcoumarin [51]. These bioactive compounds exhibit broad-spectrum activity that causes the inhibition of the growth of spoilage or pathogenic microbes associated with foods, thus suggesting their use as a natural preservative in foods [51]. The common food microflora that causes food spoilage include *Staphylococcus aureus*, *Salmonella typhimurium*, *S. typhi*, *S. enteritidis*, *Shigella dysenteriae*, *Yersinia* spp., *Penicillium expansum*, *Candida albicans*, *Aeromonas hydrophila*, *Vibrio parahaemolyticus*, *V. anguillarum*, *Aspergillus niger*, and *A. hydrophila* [52].

Additionally, the *Xylaria* spp. F0010 isolated from *Abies holophylla* has been characterized as producing griseofulvin with the chemical formula (C_17_H_17_C_l_O_6_), a spirobenzofuran antibiotic [53]. Griseofulvin has been employed for treating mycotic diseases in animals as well as human-related conditions. The in vitro and in vivo screening of genus *Xylaria* in the production of griseofulvin with microbicidal actions on plant pathogens as an effective biological agents in the control of fungal diseases affecting various food crops has been documented [54].

The extraction of Cytosporone B and C and Chaetomugilin A and D from the endophytic fungi *Chaetomimum globosum* and *Phomopsis* species inhabiting *Ginkgo biloba* with antifungal actions on *Candida albicans* and *Fusarium oxysporum* has been reported [51]. The antibacterial activities of a bioactive metabolite compound secreted by fungal endophytes in the genera *Chaetomium*, *Curvularia*, *Fusarium*, and *Aspergillus* cause inhibition growth of many bacteria [55]. The antibacterial activity (bactericidal and bacteriostatic) of *Xanthomonas compestris* and *X. oryzae* and antifungal activity of *Magnaporthe oryzae*, *Fusarium oxysporum* and *Rhizoctonia solani* isolated from *Phoma* species have been reported [56]. 

Furthermore, the antimicrobial, herbicidal and algicidal agents from chlorinated metabolites cryptosporiopsin (+) and mycorrhizin A (-) from endophytic *Pezicula* strains have been documented [57]. Similarly, the antifungal activity of chlorinated benzophenone derivatives such as Pestalachlorides A with the chemical formula (C_21_H_21_C_l2_NO_5_) and Pestalachlorides B with the chemical formula (C_20_H_18_C_l2_O_5_) extracted from the plant endophytic fungus *Pestalotiopsis adusta* against fungal plant pathogens *Verticillium albo-atrum*, *Gibberella zeae*, and *Fusarium culmorum* have been documented [58]. Kamana, et al. [59] demonstrated the antimicrobial activity of hypericin (C_30_H_16_O_8_) and emodin (C_15_H_10_O_5_) from fungal endophytes isolated from medicinal plants against *Klebsiella pneumoniae* ssp. *ozaenae*, *Pseudomonas aeruginosa*, *Staphylococcus aureus*, *Salmonella enterica*, *Escherichia coli*, *Candida albicans* and *Aspergillus niger*.

Metabolic compounds such as 2,6-dihydroxy-2-methyl-7-(prop-1E-enyl)-1-benzofuran-3(2H)-one, munumbicin A, B, C, and D, altersolanol A, 3-O-methylalaternin, phomoenamide, ambuic acid, fumigaclavine C, asperfumoid, fumitremorgin C, physcion, helvolic acid, isopestacin, ergosterol peroxide, phomodione, and pestalotheol C produced by endophytic fungi *Aspergillus fumigatus* CY018, *Pestalotiopsis theae* by various plants exhibiting potent inhibition against some fungal and viral pathogens such as *Candida albicans* are known [60].

### 3.2. Anticancer

Despite the clinical research in developing new chemotherapies, cancer remains a major disease with a high mortality rate in humans. Cancer is a human disease that causes abnormal cell growth at the point of infection that can later invade or spread to other parts of the body. In 2020, the American Cancer Society (ACS) reported 1.8 million incidences and 606,520 cancer deaths, while in 2018, World Health Organization (WHO) reported 18.1 million cancer incidence with total 9.6 million deaths. To this backdrop, a decline in the cancer incidence and mortality rates observed could be due to the intake of a balanced diet, early diagnosis and treatment. Many anticancer drugs have been synthesized and used with success. The effectiveness of anticancer drugs depends on the type, location and route of injection. The prolonged use of anticancer agents has threatened human life with numerous side effects, which include sores in the mouth and mucosa lining, hair loss, cardiac stress, nausea, bone marrow toxicity and vomiting. From this premise, evaluating the potential of some plant-associated fungi is fundamental and promising as a source of anticancer agents, but little is known about fungal anticancer agents. For example, the discovery of anticancer drug (Taxol) from an endophytic fungus, *Taxomyces andreanae* associated with the bark of *Taxus brevifolia* has been demonstrated.

As shown above, in Table 1, some examples of novel bioactive compounds from medicinal plants and endophytic fungi are listed. These compounds can be explored in the synthesis of natural products. Based on the multifunctional therapy of Taxol from *T. andreanae* that impedes the proliferation, growth and spread of cancer cells, the Food and Drug Administration (FDA) has considered it safe and approved it for treating cancer. Furthermore, studies have revealed the anticancer properties of phenylpropanoid’s amide produced from *Penicillium brasilianum* associated with the root bark of *Melia azedarach* [61]. Similarly, podophyllotoxin from *Phialocephala fortini*, *Juniperus communis*, and *Trametes hirsute* inhabiting the endosphere of *Podophyllum peltatum* and *Juniperus recurve* are known as sources of anticancer agents, respectively [62]. 

Various fungal metabolites have been produced in vivo, via in vitro assays or via fermentation technology. The screening of natural bioactive compounds from endophytic fungi has advanced their pharmacological bioprospecting in discovering new drugs. The ability of endophytic fungi to grow in the fermentation medium based on their cytotoxic actions via the secretion of specific novel bioactive metabolites has revealed their potential as an effective anticancer agent [63,64]. The potent cytotoxicity of natural-products derived from medicinal plants and their exploration is faced with a lot of challenges. Some of these challenges include low concentrations of derivable metabolites from fungal endophytes. For example, a low concentration, 0.01–0.03% of paclitaxel from the phloem of *Taxus* has been documented [65]. Furthermore, the reisolation of desired bioactive compounds from medicinal plants, geographical location, seasonal and variations in environmental conditions have been identified as serious challenges. Supply issues may also be a severe concern if a source medicinal plant is endangered or has been collected in a politically quixotic part of the world. Other problems include the destruction of essential plant crops over time since the repeated collection of plant tissues without replenishment can cause plant species to become endangered and lost after some years. 

### 3.3. Antimicrobial Compounds

#### Antitubercular

Tuberculosis (TB) is a respiratory disease that affects the human lungs, caused by a bacterium, *Mycobacterium tuberculosis*. In the world today, the prevalence of TB is commonly high among immunocompromised individuals [66]. Symptoms include fever, fatigue, weight loss, cough, night sweats, etc. According to the World Health Organization (WHO) in 2018, 1.1 million of 2.3 million new TB cases recorded affected children between ages 0–14, and 230,000 children infected with HIV-associated TB died. A further 0.83 million and 0.86 million new cases of TB were reported to be due to alcoholism and smoking, respectively.

TB is a global disease. In 2018, approximately 44% of new cases occurred in the South-East Asian region, 24% of new cases were recorded in Africa, and 18% in the Western Pacific. The most affected countries with high cases of TB include India, Nigeria, Indonesia, Pakistan, Bangladesh, South Africa, and the Philippines. The WHO reported that over 10 million people are affected by TB. One of the health targets for sustainable development goals (SDG) is to end the TB epidemic by 2030, and one of the strategies is to continue searching for nonresistant alternative antimycobacterial agents from natural sources since *Mycobacterium tuberculosis* has been known to develop resistance to many synthetic drugs. 

Most medicinal plants in our environments harbor beneficial fungal endophytes that produce active structural and biological compounds with antimicrobial activity against pathogenic microorganisms [67]. The compounds can be explored in modern medicine [68]. The isolation of fungal endophytes from various medicinal plants with the ability to synthesize new antimycobacterial drugs has been reported [21]. The bioprospecting of fungal endophytes as a remedy to tuberculosis is promising, as many fungal metabolites can easily be explored as antimycobacterials. The isolation of fungal endophytes from *Azadirachta indica* and *Parthenium hysterophorus* with antibacterial activity against tuberculosis have been reported [69]. The endophytic fungus, *Phomopsis* spp. isolated from *Garcinia* spp. that produces Phomoxanthone A and B has been reported to exhibit antimycobacterial actions against *M. tuberculosis* [51]. The 3-Nitropropionic acid and Tenuazonic acid produced by the fungal endophytes *Alternaria alternate* and *Phomopsis* spp., isolated from medicinal plants in Thailand, have reported as exhibiting potent activity against *M. tuberculosis* H37Ra by distorting isocitrate lyase enzyme pathways needed for *M. tuberculosis* metabolism and virulence [51,70]. The ability of endophytic fungus *Phomopsis* spp. isolated from *Garcinia adulcis* to synthesize the bioactive metabolites Phomoenamide and Phomonitroester has shown inhibitory effects on *M. tuberculosis* [51]. Chepkirui and Stadler [71] reported inhibition of virulent strains of *M. tuberculosis* by the bioactive compounds of benzopyran, diaportheone A and B synthesized by *Diaporthe* spp. associated with the leaves of *Pandanus amaryllifolius*. Furthermore, in vitro studies of the effects of various metabolites secreted from fungal endophytes against the etiological agents of tuberculosis have been reported [72]. 

### 3.4. Antioxidant 

Antioxidants are biological substances that prevent oxidation of chemical compounds. Some food or plant materials are rich in antioxidant content. Antioxidants protect cells from damage by reactive oxygen species (ROS) and free radicals, which cause harmful with pronounced pathological effects such as carcinogenesis, DNA damage, and degenerative diseases like Alzheimer’s disease [7]. The exploration of antioxidant components in medicinal plants is promising as an alternative therapy for treating reactive oxygen species-related diseases affecting humans [73,74]. Some of these diseases include hypertension, diabetes, cancer, atherosclerosis, cardiovascular disease, ischemia, arthritis, neurodegenerative diseases, and aging. Several antioxidant compounds from medicinal plants have been attributed to antimicrobial, anticancer, antidiabetic, anti-inflammatory, antimutagenic, antiatherosclerotic, anticarcinogenic, and antitumor properties. 

Plants and microbes are composed of polysaccharides, and studies have shown the antioxidant properties of polysaccharides obtained from living organisms. The discovery of the antioxidant activity of polysaccharides synthesized by endophytes has been reported [75]. In modern medicine, antioxidants are becoming a promising and alternative natural biological therapy for treating human diseases. The discovery of new antioxidants from plants and microorganisms in combating various diseases has been recognized as a safe and potent chemopreventive therapy for treating reactive oxygen species (ROS) associated with health conditions such as diabetes mellitus, cancer, atherosclerosis, hypertension ischemia/reperfusion injury, cardiovascular disease, rheumatoid arthritis, neurodegenerative diseases, and aging [76]. Some of the natural and new antioxidants include pestacin, corynesidones A and B, 2,14-dihydroxy-7-drimen-12,11-olide, lapachol, coumarin, 5-(hydroxymethyl)-2-furanocarboxylic acid, isopestacin, phloroglucinol, tetrahydroxy-1-methylxanthone, salidroside, *p*-tyrosol, borneol, and rutin from fungal endophytes possessing antitumor, anticarcinogenic, antimutagenic or anti-inflammatory properties [77,78,79]. These compounds with antioxidant properties are effective in the repression of damage from oxygen-derived free radicals and ROS.

Despite research conducted on antioxidants, its clinical trials are still in their infancy as few of them have been approved for clinical use; hence the need for continuing research for a novel and effective antioxidant became imperative. Studies have revealed the bioactivities of metabolites produced from several fungal endophytes. For example, the antioxidant activity of endophytic fungi genera *Xylaria* spp. and *Chaetomium* spp. isolated from *Nerium oleander* and *Ginkgo biloba* (a common medicinal plant) has been reported [80]. The antioxidants pestacin, isopestacin, graphislactone A produced by endophytic fungi *Pestalotiopsis microspore*, and *Cephalosporin* spp. isolated from *Terminalia morobensis* and *Tracheospermum jasminoides* have also been reported [78]. 

### 3.5. Antidiabetic

Diverse natural resources are present and have provided many opportunities in harnessing their potential for medicinal purposes [7]. Diabetes, referred to as *Diabetes mellitus*, is a disorder in individuals with high blood glucose (blood sugar) due to the lack of insulin production and/or the improper response of the body’s cells to insulin [81]. An increase in diabetes cases has become a significant health concern for public health professionals, thus challenging global economy and development.

The research is currently ongoing on exploring effective antidiabetic drugs from natural sources, even from microorganisms. The antidiabetic and antilipidemic activity of fungal endophytes have been reported [82]. The identification of antidiabetic peptides from the endophytic fungi *Aspergillus awamori* isolated from the medicinal plant *Acacia nilotica* using high-performance liquid chromatography (HPLC) are well documented [83]. 

Due to the side effects of some antidiabetic drugs (alpha-glucosidase inhibitor and acarbose), there is a need to search for new and alternative therapies with fewer or no harmful side effects. Fungal endophytes stand as a promising source of natural bioactive metabolites with strong antidiabetic activity and have been considered viable and economical [60]. In vivo and in vitro assays of compound lectin (N-acetylgalactosamine, 64 kDa) produced by the endophytic fungus *Alternaria* spp. isolated from *Viscum album* with healthy antidiabetic activity in the diabetic rat are known [84]. The nonpeptidal metabolite demethylasterriquinone B-1 (L-783,281) possessing insulinlike activity produced by endophytic fungus *Pseudomassaria* spp. isolated from African rainforest has also been reported [5]. 

The antidiabetic potential of endophytic fungal endophytes has indicated their possibility as a source of antidiabetic compounds. The reduction in the fasting blood sugar of experimented diabetic mice administered with purified compounds such as (S)-(+)-2-cis-4-trans-abscisic acid, 7′-hydroxy-abscisic acid, and 4-des-hydroxyl altersolanol A from an endophytic fungus *Nigrospora oryzae* associated with the leaves of *Combretum dolichopetalum* has been reported [85]. Indrianingsih and Tachibana [86] also reported strong α-glucosidase inhibitory activity on 8-hydroxy-6,7-dimethoxy-3-methyl isocoumarins, a chemical compound produced by an endophytic fungus, *Xylariaceae* spp., from the stem of *Quercus gilva* Blume. 

The antidiabetic and hypolipidemic activity extracts from *Phoma* spp. and *Aspergillus* spp. isolated from *Salvadora oleoides* in Wistar albino rats when orally administered glucose and alloxan have been reported to reduce blood sugar levels in rats [87]. The assessment of endophytic fungi from two prominent medicinal plants *Rauwolfia densiflora* and *Leucas ciliate*, with antidiabetic bioprospecting for treating diabetes is known [7,88]. The authors further reported that the α-amylase inhibitor significantly reduces glucose from the complex carbohydrates and slows down the absorption rate of glucose. The screening of endophytic fungi as a precursor for alpha-glucosidase inhibitors has been reported [89]. The antidiabetic activity of compounds produced from *Fusarium* spp. and *Alternaria* spp., as a precursor to alpha-glucosidase inhibitors, has been reported in the literature [7], thus, establishing the multifunctional prospect of some fungal endophytes as a source of pharmaceuticals. 

### 3.6. Antiparasitic and Antimalarial

Parasitism is a form of symbiotic association that occurs between organisms with one benefiting and other being harmed. A parasite is referred to as an organism that feeds in/on the host and causes damage. Parasites live in/on the host cell [90]. Different microorganisms exist parasitically. Protozoans or helminths are grouped as pathogenic parasites, and their presence in the host plant induced harmful effects. The etiological agent of malaria is *Plasmodium* species. The common *Plasmodium* spp. include *Plasmodium vivax*, *P. malariae*, *P. ovale*, and *P. falciparum*. Globally, the transmission of this protozoan among individuals has caused more than 3.3 billion deaths [7,42]. In 2016, approximately 228 million new malaria cases with more than 405,000 deaths were reported in 91 countries. Malaria is prevalent among individuals in the tropical and Sub-tropical regions of the world; people living in sub-Saharan Africa and Southeast Asia were most affected, with more than 80% cases caused by *Plasmodium falciparum* [90]. This parasite affects approximately 45% of the world’s population, especially in developing countries, due to inadequate or insufficient health facilities. Recent reports on drug resistance by malaria parasites have led to an urgent need for alternatives and effective antimalarial drugs with endophytic fungi as promising bioprospecting candidates. Therefore, there is a need to explore bioactive metabolites possessing antimalarial features. 

Fungal endophytes have great potential in the synthesis of new antimalarial drugs in the pharmaceutical industry. The synthesis of bioactive metabolite phomoxanthones A-C, an aromatic sesquiterpene, by *Phomopsis archeri* isolated from *Vanilla albindia* with remarkable antimalarial properties has been reported [91]. The protozoan genera *Trypanosoma* and *Leishmania* are examples of pathogenic parasites. Wang, et al. [92] reported the inhibition of the growth of *Leishmania* by the bioactive compound Cochlioquonone A produced by *Cochliobolus* spp. isolated from *Piptadonia adiantiodes*. The Cercosporin synthesized by *Mycosphaerella* spp. from *Psychotoria horizontalis* has also been effectively employed as an antiparasitic agent against *Trypanosoma cruzi*, *Plasmodium falciparum* and *Leishmania donovani* [92].

The antiplasmodial activity of some endophytic fungi producing certain metabolites against *Plasmodium falciparum* has been reported [93,94]. The antimalarial activity of the endophytic fungus *Diaporthe miriciae* producing the bioactive metabolite epoxycytochalasin H against *Plasmodium falciparum*-resistant strains to antibiotics (chloroquine) has been reported [95]. The in vitro antiplasmodial activity and phytotoxicity of 19,20-epoxycytochalasins C and D, cytochalasins and 18-deoxy-19,20-epoxy-cytochalasin C, produced by *Nemania* spp. UM10M isolated from the leaf of diseased *Torreya taxifolia* has been reported [96]. Similarly, the research performed by Ateba et al. [42] on the potent antiplasmodial extracts from endophytic fungi isolated from *Symphonia globulifera* against an antibiotic chloroquine-resistant strain *Plasmodium falciparum* (PfINDO) revealed the potential of isolated endophytic fungi as curative agents for treating malaria. Toghueo, et al. [97] also reported the antiplasmodial activity of *Aspergillus versicolor* AMb7, *Trichoderma afroharziamun* AMrb7, *Neocosmospora rubicola* AMb22, *Penicillium citrillium* AMrb11, *P. citrillium* AMrb23, and *Fusarium* spp. AMst1 against *P. falciparum* strain.

### 3.7. Antiviral

To find a lasting solution to antibiotic resistance tendencies in microorganisms, developing new antiviral drugs is required with utmost urgency. The bioactive metabolites from endophytic fungi stand a chance as suitable candidates for the synthesis of antiviral agents, thus making this a fascinating area of study. The bioprospecting of endophytic fungi for the synthesis of antiviral agents is promising, although little information has been documented on their exploration. The major constraints encountered in the discovery of antiviral compounds are attributed to inefficient and inappropriate or absent antiviral screening measures in most metabolite compound discovery programs. However, antiviral agents, including cyclosporine U, cytonic acid A, and B, S39163/F-I, podophyllotoxin, sequoiatones C-F, and CR377 have been reported from some fungal endophytes [5].

Antiviral compounds from endophytic fungi possessing strong activity against some viruses, which include human immunodeficiency virus (HIV) [98], human cytomegalovirus, Dengue virus [99], and influenza A (HINI) virus [100] have been reported. The antiviral properties of two new compounds, cytonic acid A (C_32_H_36_O_10_) and B (C_32_H_36_O_10_) isolated from *Cytonaema* spp. have been reported [51]. The structural elucidation of p-trideside isomers by mass spectrometry and NMR methods has revealed novel human cytomegalovirus protease inhibitors. The antiviral compound Hinnuliquinone produced by fungal endophytes colonizing the plyllosphere (leaves) part of an oak tree (*Quercus coccifera*) has been attributed as inhibitor against HIV-1 protease [51,68]. The pullularins A-D (cyclo hexadepsi-peptides) produced by *Pullularia* spp. BCC 8613 were found to exhibit antimalarial activities against *P. falciparum* and antiviral activities against herpes simplex virus (HSV) [51]. The anti-HIV properties of the antiviral compound Pestalotheol-C produced by *Pestalotiopsis theae* from an unidentified tree on Jianfeng Mountain, China have been reported [51].

The endophytic fungus *Alternaria tenuissima* QUE1Se produces Altertoxins, an effective compound against HIV-1 virus. Several compounds produced from *Emericella* spp. (HK-ZJ), which include emerimidine (A, B), dehydroaustin, austinol, include aspernidine (A, B), Austin, emeriphenolicins (A, D) and acetoxy dehydroaustin have been reported to exhibit antiviral activity against influenza A virus (H1N1) [99]. The crude extract from most medicinal plants has displayed a high degree of antiviral activity. Besides the fungal endophytes found in the plants, some groups of actinomycetes have been reported to display antiviral activities [7]. For example, *Streptomyces* spp. GT2002/1503 has been reported to exhibit antiviral actions against R5 tropic HIV infection [100]. The antiviral compound 2-(furan-2-yl)-6-(2S, 3S, 4-trihydroxybutyl) pyrazine produced from *Jishengella endophytica* 161111 with antiviral activity against influenza A virus (subtype H1N1) has been reported [7].

### 3.8. Immunosuppressive

The roles of fungal endophytes are known in agriculture and in the pharmaceutical industry [101]. Nowadays, the search for endophytic resources in clinical medicine is ongoing as a source of immunomodulatory compounds with a prospect for treating autoimmune disorders such as insulin-dependent diabetes and rheumatoid arthritis and precursors that avert allograft declination in transplant patients [102]. Fungal endophytes are capable of synthesizing certain compounds with immunosuppressive action [103]. Due to the heterogeneous mechanisms of most of the synthetic (chemical) immunosuppressive drugs, their effects are consequentially challenging due to their prolonged use for treating diseases. The side effects of the continuous use of chemical immunosuppressive drugs include hyperlipidemia, nephrotoxicity, hypertension, neurotoxicity, and the risk of infection [104].

The continuous intervention in the search for effective and friendly immunosuppressive drugs is currently ongoing and recent research findings have discovered potent immunosuppressive therapies from fungal endophytes including sydoxanthone A and B, colutellin A, 13-O-acetylsydowinin B, dibenzofurane, methyl peniphenone, xanthone derivatives, subglutinol A and B, lipopeptide, peniphenone, benzophenone derivatives, (-) mycousnine, polyketide benzannulated spiroketal, and polyketide benzannulated spiroketal [105]. 

The isolation of the immunosuppressant drug Cyclosporin A from *Tolypocladium inflatum* has been reported [106]. The finding of cyclosporin A extracted from the soil endophytic fungus *Trichoderma polysporum* as a principal immunosuppressive agent is also known. Diterpene pyrenes, a noncytotoxic compound, and Subglutinol A and B produced by fungal endophytes *Fusarium subglutinan* isolated from *Tripterygium wilfordii* with immunosuppressive action have also been reported [107]. The inclusion of Subglutinol A and B in an equal amounts of 0.1 μM to mixed lymphocyte reaction assay and thymocyte proliferation assay yielded an equal percentage (50%) inhibitory concentration [108]. Similarly, the commercial immunosuppressant drug cyclosporine used in the same assay systems has shown inhibition in the mixed lymphocyte reaction assay and 10^4^ more in the thymocyte proliferation assay. Nevertheless, the shortfall in the toxicity associated with subglutinols A and B necessitates possible exploration in further studies. The cyclosporine produced by *Tolypocladium inflatum* can be a potential candidate for large exploration of immunosuppressants. The example given by the authors has depicted the progression of many scientists in the search for unknown or unidentified endophytes from fascinating and uncommon hosts and environments.

The synthesis of mycophenolic acid from endophytic fungi in the genera *Busssochlamys*, *Penicillium*, *Septoria*, and *Aspergillus* has been reported [109]. Mycophenolic acid is a potent immunosuppressant used in medicine for treating autoimmune diseases, thus preventing rejection reaction in a transplanted organ. The in vivo and in vitro studies conducted by Lin, et al. [110] reported the effectiveness of subglutinol A causing blockages in the T-lymphocytes’ (T-cell) growth and survival. The inhibition of colutellin A on CD_4_ (cluster differences 4)—T cell activation of IL-2 (Interleukin 2) produced with an IC_50_ (half maximal inhibitory concentration) of 167.3/0.38 nM with nontoxicity on human peripheral blood mononuclear cells has been reported [111]. The potential of natural bioactive compounds subglutinol A and colutellin A produced from endophytic fungi can be used as alternative immunosuppressive drugs for treating autoimmune diseases. However, the identification of actual molecular specificity and explicit mechanisms of action of these drugs is unknown. The novel compounds from fungal endophytes can accomplish the recent demand for new and affordable immunosuppressive therapeutic drugs that can be explored as medicaments to autoimmune diseases and posttransplantation care.

## 4. Conclusions and Future Outlook 

Research into the microbial world of plants is currently fascinating with great promise in agriculture, pharmaceuticals and medicine. The efficacy of medicinal plants and associated endophytes has gained attention from scientists in the assessment of their pharmacological potential in the synthesis of the bioactive compounds naturally present in them. Medicinal plants contain many phytochemical constituents that can be explored as medicaments to human diseases. 

Currently, the outlook of medicinal compounds and high demand for nontoxic drugs have diversified in the search for novel biologically active metabolites from plant endophytes. Although the exploration of bioactive compounds and the use of some medicinal plants such as sunflower in traditional medicine has long been studied, little information is available on the bioactivities of its associated fungal endophytes. Furthermore, the research into various pharmacological effects of bioactive compounds from endophytes is still in infancy. Nevertheless, review documentation on their functions can provide insights into possible explorations, as outlined in this review work.

Thus, the screening of fungal endophytes for possible bioactive metabolite synthesis can help in the establishment of their pharmacological functions as well as for research in providing information on the bioprospecting of endophytic fungi as biological entities that will ensure sustainable human health and action against antibiotic resistance. 

## Figures and Tables

**Table 1 jof-07-00147-t001:** The effects of bioactive compounds of fungal endophytes associated with medicinal plants.

Medicinal Plant	Endophytic Fungi	Bioactive Compounds/Effects	References
*Taxus brevifolia* *Aegle marmelos* *Plectranthus amboinicus*, *Wollemia nobilis* *Ginkgo biloba* *Taxus media* *Taxodium distichum* *Terminalia arjuna* *Citrus medica*	*Taxomyces andreanae* *Bartalalinia robillardoide* *Pestalotiopsis microspore* *P. guepinii* *P. microspore* *P. terminaline* *Cladosporium cladosporio* *Phyllosticta citricarpa*	Anticancer/Antitumor Paclitaxel	[30,31]
*Taxodium distichum* *Rhizophora annamalayana* *Taxus baccata*	*Alternaria alternata*, *Aspergillus fumigatus* *Pestalotiopsis terminaliae*, *Wollemia nobilis*, *Baralinia robillardoides*, *Taxodium distichum*, *Phyllosticta spinarum*, *Botrydiplodia theobromae*	Taxol	
*Taxus baccata* *Camptotheca acuminate* *Apodytes dimidiate*	*Fusarium oxysporum*, *Fusarium redolens*, *Fusarium solani*, *Trichoderma atroviride*	Camptothecin	
*Sinopodophyllum hexandrum* *Diphylleia sinensis* *Adenophora axilliflora*	*Fusarium solani* *Penicillium implicatum* *Mucor fragilis*, *Phialocephala fortinii*	Podophyllotoxin Chaetominine	
*Melia azedarach* *T. taxifolia* *D. cejpii*	*Penicilium brasilianum* *Pestalotiopsis microspore* *Dichotomomyces albus*	Phenylpropanoids Torreyanic acid Xanthocillin X	
*Cardiospermum helicacabum* *Nothapodyte foetida* *Juniperus communis* *Juniperus recurva* *Podophyllum hexandrum* *Dysosma veitchii Podophyllum peltatum* *Sinopodophyllum hexandrum*	*Pestalotiopsis pauciseta* *Entrophospora infrequent* *Aspergillus fumigatus* *Fusarium oxysporum* *Trametes hirsute* *Penicillium implicatum* *Phialocephala fortinii* *Alternaria neesex*	Camptothecin Podophyllotoxin	
*Catharanthus roseus* *Tripterygium wilfordii* *Roystonea regia* *Cyndon dactlon*	*Fusarium oxysporum* *Rhinocladiella* spp. *Pestalotiopsis photiniae* *Aspergillus niger* IFB-E003	Vincristine Cytochalasins Photinides Rubrofusarin B	
*Urospernum picroide* *Garcinia dulcis* *Saurauia scaberrinae* *Torreya taxifolia* *Kennedia nigriscans* *Crytosporiopsis cf quercina*	*Ampelomyces* spp. *Phomopsis* spp. PSU-D15 *Phoma* spp. *Pestalotiopsis microspora* *Streptomyces* NRRL 30562 *Cryptosporiopsis* spp.	Antimicrobial 3-0 methylalaternin, altersolanol A Phomoenamides Phomodione Ambuic acid Mumubicin Cryptocandin	[32]
*Sabina recurve* *-*	*Fusarium oxysporum* *Penicillium chrysogenum*	Antiviral Cyclosporine Xanthoviridicatins	[30]
*Smallanthus sonchifolius* *Viguiera arenaria*	*Curvularia* spp. *Phomopsis* spp.	Antimalaria/antiparasite Stemphyperylenol 3,4-dimethyl-2-(40-hydroxy-30,50-dimethoxyphenyl)-5-methoxy-tetrahydrofuran	[28]
*Citrus limon*	*Penicillium digitatum*, *P. citrinum*	Antifungal Tryptoquialanine A, Tryptoquialanine C, 15-dimethyl-2-epi-fumiquinazoline A, deoxytryptoquialanone, Citrinadin A, Deoxycitrinadin A, Chrysogenamide A	[33,34]
*Senecio kleinii*	*Phoma* sp.	8,9-dihydro-3,5,7-trihydroxy-1,8,8,9-tetramethyl-5-(2-oxopropyl)-4H-phenaleno[1,2-b]furan-4,6(5H)-dione, atrovenetinone, sclerodione	[35]
*Acacia nilotica* *Adhatoda beddomei*, *Ficus religiosa*, *Paeonia delavayi*, *Salvadora oleoides Decne* *Sonneratia ovata*	*Aspergillus awamori* *Syncephalastrum* spp. *Dendryphion nanum*, *Phomopsis* spp. *Aspergillus* spp *Phoma* spp. *Nectria* spp.	Antidiabetic Peptide lectin (N-acetylgalactosamine, 64 kDa) Naphthoquinones (O-phenethylherbarin), herbarin and herbaridine, phomopoxides A-G, 2,6-di-tert-butyl-p-cresol, phenol-2,6-bis[1,1-dimethylethyl]-4-methyl Citreoisocoumarinol, citreoisocoumarin, and macrocarpon C	
*Solanum xanthocarpum*	*Aspergillus terreus*, *A. sclerotiorum*, *A. terreus*	Antioxidants Lovastatin	[36]
*Magnifera casturi*, *Cestrum nocturnum*, *Nerium oleander*, *Saposhnikovia**divaricata*, *Acalypha indica*, *Azadirachta indica*, *Catharanthus roseus*, *Pediomelum cuspidatum*, *Artemisia capillaris*, *Catharanthus roseus* *Caralluma acutangula*, *Rhazya stricta*, and *Moringa peregrina* *Eugenia jambolana*, *Fritillaria unibracteata*, *Gymnema sylvestre*, *Kandis gajah*	*Aspergillusminisclerotigenes*, *Asper. oryzae*, *Asper. wentii*, *Rhodiola crenulata*, *Rhod. angusta*, *Rhod. sachalinensis*, *Chaetomium* sp., *Diaporthe phaseolorum*, *Colletotrichum kohawae*, *Phomopsis phylanthicolla*, *Xylaria* acuta, *Alternaria alternata*, *Bipolaris sorokiniana*, and *Cladosporium sphaerosperumum* *Aspergillus niger*, *Aspergillus peyronelii*, *Aspergillus sp*., *and Chaetomium globosum*, *Fusarium tricinctum*, *Clonostachys rosea*, *Gymnema sylvestre*, *Chrysonilia sitophila* *Alternaria alternata*	Phenolic acids (Chlorogenic acid (5-O-caffeoylquinic acid), Di-O-caffeoylquinic acids Flavonoids (Quercetin 3-rutinoside) (rutin), Quercetin 3-rhamnoside, (quercitrin) Quinones (Anthraquione glycoside) Rehein, emodin Volatile compounds (Artemisin) Aliphatic compounds Hexadecanoic acid, methyl ester; 9-Hexadecenoic acid, methyl ester; 9,12 Octadecadienoic acid, methyl ester; 11,14,17-Eicosatrienoic acid, methyl ester DPPH (2,2-diphenyl-1-picrylhydrazyl) 3,5-dihydroxy-2,5-dimethyltrideca 2,9,11-triene- 4,8-dione Chrysin	[34,37,38,39]
*Polygonum cuspidatum*	*Streptomyces* spp.	3-methyl-1-butanol, 4-methyl-1-pentanol, 1-nonanal, 6-methyl-2-oxiranyl-hept-5-en-2-ol, 2,6,11,15-tetramethylhexadecane, 2,6-dimethylocta-2, 7-dien-6-ol, 2,4-di-tert-butylphenol, glacial acetic acid, linoleic acid, 4-methylvaleric acid, 4-hexenoic acid, dehydroacetic acid, heptanedioic acid, 2-methyl butyric acid, and 1-p-menthen-8-ol	[40]

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
