# Peer review of "Pharmacological Potential of Fungal Endophytes Associated with Medicinal Plants: A Review"

_jof, 2021, doi:10.3390/jof7020147_

Round 1

Reviewer 1 Report

The paper is well structured and well written, but there are some concerns that need to

The paper is well structured and well written, but there are some concerns that need to be reviewed before publication.

  • it should be better isert  rows and columns into table 2.
  • check the species name, they should be written in italics, for example lines 221-222.

Author Response

The Editorial Officer (Ms. Lily Guo),
I am pleased to submit our revised manuscript titled: ‘Pharmacological potential of fungal endophytes associated with medicinal plants: a review’. All reviewer comments are well appreciated. We have taken our time to critically attend to each of the comments in the main body of the manuscript. Our response to each of the points raised is presented below:

Comments to the author (if any):

Response: All the comments raised by the reviewers have been carefully checked and revised appropriately.

Reviewer 1

Comments and Suggestions for Authors

The paper is well structured and well written, but there are some concerns that need to

The paper is well structured and well written, but there are some concerns that need to be reviewed before publication.

  • it should be better insert  rows and columns into table 1.
  • Response: The rows and columns have been inserted in Table 1 (Page 3-5).
  • check the species name, they should be written in italics, for example lines 221-222.
  • Response: The species name has been italicized and revised appropriately. Where corrections needed to be made has been taken to Table 1 in Taxol column (Page 3-5).

Reviewer 2 Report

General comment: Some information on mechanisms of action, effective concentrations and animal studies, if any should be discussed in each section. 

Lines 28-29: Despite the tiny nature of fungi, they serve as an invaluable source of novel metabolite compounds and……

Comment: The size of fungi is not an issue. The authors should state that fungi have an extensive repertoire of secondary metabolites.

Lines 43-44: host plants to survive under different environmental stresses by secretion of notable metabolites

Comment: “notable” should be replaced by “bioactive” or “stress-adaptor”

Lines 61-62:  compounds has mirrored similar antimicrobial properties on the associated endophytic fungi

Comment: if it has been proved that the antimicrobial properties derive from the endophytes this should be stated. If this is unclear, then the authors must state “it is considered that the bioactive compounds are synthesized by the endophytic fungi, but this has not been demonstrated”.

Lines 88-89: Research has established 88 that most plant-derived metabolites can as well be derived from its associated fungi

Comment: this is not accurate, the authors should give an estimate of what % of plant metabolites can be produced by the associated fungi.

Lines 97-102: Endophytic fungi potentially produce an analog of novel bioactive compounds. To maximally explore, suitable media and growth parameters need to be adjusted for biomass yield. Limiting nutrients in the growth media influence the quantity and efficacy of bioactive compounds. An insight into fungal studies would be advantageous in deriving novel pharmaceutical products.

Comment: This part should be rewritten as “Endophytic fungi potentially produce novel bioactive compounds. Suitable media, growth parameters and nutrient limitation should be explored to gain an insight into fungal metabolism in order to discover novel pharmaceutical products”.

Comment for Line 105: The section 3.1 should be titled antifungals.

Lines 108-110:   anda limited number of antimicrobial production [33]. Currently, the plant disease control and health status of humans via the use or application of pharmaceutical drugs is of major concern.

Comment: Currently, the major concerns of antimicrobial drugs are for plant disease and human health.  Please check grammar and typos.

Comment for Lines 121-122:  Table 1 should come before Table 1. Please fix the Table numbers. In Table 1 (Table 2 currently), the authors should justify why antitumor and anticancer are shown as exclusive categories. A category for antifungals whether targeting human pathogens or plant pathogens/food spoilage fungi should be shown since there are many examples in the text.   

Lines 133-134: Antimicrobial compounds are relatively low in molecular weight, and even at a low concentration, they exhibit high activity against pathogenic microorganisms.

Comment: The authors should mention a concentration range instead of just “low”.

Comments for the anticancer section: This section needs to be organized into well-known substances, newer compounds, their sources and challenges in their development. There is a lot of redundant information that needs editing.

Section 3.3 should be titled “antimicrobial”; if anti-tubercular drugs appear, they have to be a sub-section under antimicrobial. In addition, the sections antidiabetic and antioxidant categories are not mentioned in the table, which should probably be divided into infectious and non-infectious diseases. 

Line 383: The antidiabetic activity of Fusarium spp and Alternaria spp

Comment: The mentioned fungi are not the precursors, rather the compounds produced from them are.      

Author Response

The Editorial Officer (Ms. Lily Guo),
I am pleased to submit our revised manuscript titled: ‘Pharmacological potential of fungal endophytes associated with medicinal plants: a review’. All reviewer comments are well appreciated. We have taken our time to critically attend to each of the comments in the main body of the manuscript. Our response to each of the points raised is presented below:

Comments to the author (if any):

Response: All the comments raised by the reviewers have been carefully checked and revised appropriately.

Reviewer 2

Comments and Suggestions for Authors

General comment: Some information on mechanisms of action, effective concentrations and animal studies, if any should be discussed in each section. 

  • Response: Additional comments have been added to each section where appropriate.

Lines 28-29: Despite the tiny nature of fungi, they serve as an invaluable source of novel metabolite compounds and……

Comment: The size of fungi is not an issue. The authors should state that fungi have an extensive repertoire of secondary metabolites.

  • Response: The fungus “Penicillium notatum” that has extensive repertoire of secondary metabolites is clearly stated and revised (Page 1).

 Lines 43-44: host plants to survive under different environmental stresses by secretion of notable metabolites

Comment: “notable” should be replaced by “bioactive” or “stress-adaptor”

  • Response: “notable” has been replaced by “stress-adaptor” as suggested by the reviewer (Page 2).

Lines 61-62:  compounds has mirrored similar antimicrobial properties on the associated endophytic fungi

Comment: if it has been proved that the antimicrobial properties derive from the endophytes this should be stated. If this is unclear, then the authors must state “it is considered that the bioactive compounds are synthesized by the endophytic fungi, but this has not been demonstrated”.

  • Response: The sentence “it is considered that the bioactive compounds are synthesized by the endophytic fungi, but this has not been demonstrated” as suggested by the reviewer has been added and modify (Page 2).

 Lines 88-89: Research has established 88 that most plant-derived metabolites can as well be derived from its associated fungi

Comment: this is not accurate, the authors should give an estimate of what % of plant metabolites can be produced by the associated fungi.

  • Response: The % of plant metabolites produced by the associated fungi “approximately 18%” has been included in the body of the manuscript (Page 2).

 Lines 97-102: Endophytic fungi potentially produce an analog of novel bioactive compounds. To maximally explore, suitable media and growth parameters need to be adjusted for biomass yield. Limiting nutrients in the growth media influence the quantity and efficacy of bioactive compounds. An insight into fungal studies would be advantageous in deriving novel pharmaceutical products.

Comment: This part should be rewritten as “Endophytic fungi potentially produce novel bioactive compounds. Suitable media, growth parameters and nutrient limitation should be explored to gain an insight into fungal metabolism in order to discover novel pharmaceutical products”.

  • Response: The part as suggested by the reviewer has been rewritten as “Endophytic fungi potentially produce novel bioactive compounds. Suitable media, growth parameters and nutrient limitation should be explored to gain an insight into fungal metabolism in order to discover novel pharmaceutical products” (Page 3).

 Comment for Line 105: The section 3.1 should be titled antifungals.

  • Response: The title is corrected and revised to “antifungal effects” (Page 6).

Lines 108-110:   anda limited number of antimicrobial production [33]. Currently, the plant disease control and health status of humans via the use or application of pharmaceutical drugs is of major concern.

Comment: Currently, the major concerns of antimicrobial drugs are for plant disease and human health.  Please check grammar and typos.

  • Response: The typos has been corrected to “and a” while the confusing grammar has been expunged from the body of the manuscript (Page 6).

Comment for Lines 121-122:  Table 1 should come before Table 1. Please fix the Table numbers. In Table 1 (Table 2 currently), the authors should justify why antitumor and anticancer are shown as exclusive categories. A category for antifungals whether targeting human pathogens or plant pathogens/food spoilage fungi should be shown since there are many examples in the text.   

  • Response: The table number has been corrected as “Table 1”. The antimuor/anticancer were corrected and revised accordingly. Category of antifungals has has been included in Table 1 (Page 3-5).

Lines 133-134: Antimicrobial compounds are relatively low in molecular weight, and even at a low concentration, they exhibit high activity against pathogenic microorganisms.

Comment: The authors should mention a concentration range instead of just “low”.

  • Response: The concentration range of (0.5 - 8.0 mg/ml) has been included and referenced in the body of the manuscript (Page 6).

Comments for the anticancer section: This section needs to be organized into well-known substances, newer compounds, their sources and challenges in their development. There is a lot of redundant information that needs editing.

  • Response: The anticancer section has been critically revised and organized. Example of some new compounds and their sources were highlighted. The challenges in their development is well structured and listed (Page 7-8).

Section 3.3 should be titled “antimicrobial”; if anti-tubercular drugs appear, they have to be a sub-section under antimicrobial. In addition, the sections antidiabetic and antioxidant categories are not mentioned in the table, which should probably be divided into infectious and non-infectious diseases.

  • Response: Section 3.3 has been titled “antimicrobial” with subheading “anti-tubercular” (Page 8) while the antidiabetic and antioxidant have been included in Table 1 (Page 4-5).

Line 383: The antidiabetic activity of Fusarium spp and Alternaria spp

Comment: The mentioned fungi are not the precursors, rather the compounds produced from them are.      

  • Response: The sentence has been corrected and revised appropriately in the body of the manuscript (Page 10).

    Thank you for taking the time to review our manuscript. We hope for your kind response soon.

    Thank you.
    Professor Olubukola Oluranti Babalola.
    27 January 2021.

Round 2

Reviewer 2 Report

Line 371: antimalarial should be changed to antiparasitic. 

Line 407-408: should be changed to antimalarial activities against P. falciparum & antiviral activities against HSV. 

Author Response

Comments to the author (if any):

Response: All the comments and suggestions raised by reviewer 2 is well appreciated and have been carefully checked and revised appropriately.

Reviewer 2

Comments and Suggestions for Authors

Line 371: antimalarial should be changed to antiparasitic. 

  • Response: The word “antimalarial” in Line 371 as suggested by the reviewer has been changed and revised to “antiparasitic” in the body of the manuscript (Page 10).

Line 407-408: should be changed to antimalarial activities against P. falciparum & antiviral activities against HSV.

  • Response: The sentence in line 407-408 has been revised accordingly in the body of the manuscript to “antimalarial activities against falciparum and antiviral activities against herpes simplex virus (HSV)” (Page 11).

Thank you for taking the time to review our manuscript.